The reanalysis of biogeography of the Asian tree frog, Rhacophorus (Anura: Rhacophoridae): geographic shifts and climatic change influenced the dispersal process and diversification

Pan Tao 1
Zhang Yanan 1
Wang Hui 1
Wu Jun 2
Kang Xing 1
Qian Lifu 1
Chen Jinyun 3
Rao Dingqi 4
Jiang Jianping 5
Zhang Baowei zhangbw@ahu.edu.cn 1 6
1 Anhui Key Laboratory of Eco-engineering and Bio-technique, School of Life Sciences, Anhui University , Hefei , Anhui Province , China
2 Ministry of Environmental Protection, Nanjing Institute of Environmental Sciences , Nanjing , Jiangsu , China
3 Department of Life Science, Huainan Normal University , Huainan , Anhui , China
4 Kunming Institute of Zoology, Chinese Academy of Sciences , Kunming , Yunnan , China
5 Chengdu Institute of Biology, Chinese Academy of Sciences , Chengdu , China
6 School of Biosciences, Cardiff University , Cardiff , United Kingdom
Pie Marcio
Electronic publication date: 2017 Nov 21
Publication date: 2017
Volume: 5
Electronic Location ID: e3995
Received 2017 Aug 15; Accepted 2017 Oct 16
Copyright: ©2017 Pan et al.
Copyright year: 2017
Copyright holder: Pan et al.
License: This is an open access article distributed under the terms of the Creative Commons Attribution License, which permits unrestricted use, distribution, reproduction and adaptation in any medium and for any purpose provided that it is properly attributed. For attribution, the original author(s), title, publication source (PeerJ) and either DOI or URL of the article must be cited.
License URL: https://creativecommons.org/licenses/by/4.0/

Keywords: Dispersal process, Rhacophorus, Diversification, Geographic shift, Climate change

Funding: National Natural Science Foundation of China 31272332 31071894 30911120031 30670243 Anhui Province Higher Education Revitalization Plan 2014 Colleges and Universities Outstanding Youth Talent Support Program This work was supported by the National Natural Science Foundation of China (Grant No. 31272332, 31071894, 30911120031, 30670243), Anhui Province Higher Education Revitalization Plan, 2014 Colleges and Universities Outstanding Youth Talent Support Program. The funders had no role in study design, data collection and analysis, decision to publish, or preparation of the manuscript.

==============================
Rapid uplifts of the Tibetan Plateau and climate change in Asia are thought to have profoundly modulated the diversification of most of the species distributed throughout Asia. The ranoid tree frog genus Rhacophorus, the largest genus in the Rhacophoridae, is widely distributed in Asia and especially speciose in the areas south and east of the Tibetan Plateau. Here, we infer phylogenetic relationships among species and estimate divergence times, asking whether the spatiotemporal characteristics of diversification within Rhacophorus were related to rapid uplifts of the Tibetan Plateau and concomitant climate change. Phylogenetic analysis recovered distinct lineage structures in Rhacophorus, which indicated a clear distribution pattern from Southeast Asia toward East Asia and India. Molecular dating suggests that the first split within the genus date back to the Middle Oligocene (approx. 30 Ma). The Rhacophorus lineage through time (LTT) showed that there were periods of increased speciation rate: 14–12 Ma and 10–4 Ma. In addition, ancestral area reconstructions supported Southeast Asia as the ancestral area of Rhacophorus. According to the results of molecular dating, ancestral area reconstructions and LTT we think the geographic shifts, the staged rapid rises of the Tibetan Plateau with parallel climatic changes and reinforcement of the Asian monsoons (15 Ma, 8 Ma and 4–3 Ma), possibly prompted a burst of diversification in Rhacophorus.

Introduction

Abiotic factors like climatic and tectonic events, and biotic factors like inter- or intraspecific interactions, competition and predation may be the predominant driving factors during the evolution and diversification of organisms (Antonelli & Sanmartín, 2011; Benton, 2009). These factors can also affect the diversification at different temporal and geographical scales (Benton, 2009). Understanding the processes of diversification and which factors have driven the evolution and diversification of organisms, may help guide prioritization in conservation and forecast the population demography under future climate conditions (Avise, 2000; Frankham, Briscoe & Ballou, 2002). It had been shown that orogenic activity during recent geological history was linked to the formation of hotspots of biodiversity (Myers et al., 2000). Consequently, the study of the origin and evolution of biodiversity in mountain systems has experienced a growing scientific interest (Klaus et al., 2016; Van Der Meijden et al., 2007; Zhang et al., 2006). In Asia, the uplift of the Tibetan Plateau was the most intense orogenic movement leading to the formation of several biomes (Favre et al., 2015; Klaus et al., 2016; Myers et al., 2000; Yang, Dong & Lei, 2009). Therefore, research has focused on the uplift of the Tibetan Plateau, based on a temporal (molecular dating) and spatial (biogeographic) framework, which may have triggered a series of evolutionary changes in different biological groups (Klaus et al., 2016), such as in plants (Gao et al., 2013; Jabbour & Renner, 2012; Tu et al., 2010; Wang et al., 2009), birds (Lei, Qu & Song, 2014; Tietze & Borthakur, 2012; Tietze et al., 2013), mammals (Deng et al., 2011) and amphibians (Che et al., 2010; Guo et al., 2011; Li et al., 2013; Vijayakumar et al., 2016; Zhang et al., 2006).

The ranoid treefrog genus, Rhacophorus, is the largest genus in the Rhacophoridae, currently containing 88 species (Frost, 2016), which are widely distributed across India, China, Japan, mainland South-east Asia, the Greater Sunda Islands and the Philippines (Frost, 2016). A previous study disclosed that Rhacophoridae underwent an early dispersal from India to Asia between 46 and 57 Ma, that a transient faunal exchange ceased during the Middle Eocene, and a subsequent increase of Rhacophorid dispersal events between Asia and the Indian subcontinent during the Oligocene that continued until the Middle Miocene (Li et al., 2013). Uplift of the Tibetan Plateau and a series of climatic and environmental changes led to many speciation events on a very large scale (Favre et al., 2015; Myers et al., 2000; Yang, Dong & Lei, 2009). Rhacophorus taxa are widely distributed across the areas around the Tibetan Plateau, according to previous study (Li et al., 2013), the speciation process in this genus may be linked to the uplift of Tibetan Plateau during the Miocene and Pliocene.

To gain a better understanding of the diversification processes in biomes around the Tibetan Plateau, we herein provide a historical biogeographic pattern of Rhacophorus. In the present study, we collected all the sequences datasets of Asian Rhacophorus that have been reported in addition to newly sequenced DNA from Rhacophorus specimens collected from the Dabie Mountains in Anhui, China. We infer the phylogenetic relationships within the genus and estimate the divergence times. Further, the correlation between diversification events within Rhacophorus and the geographic shifts in the Tibetan Plateau are explored.

Materials and Methods

Ethical statement

The collection of samples was performed within a long-term investigation project on amphibians of Dabie mountains. This investigation project and the sample collection were approved by the Animal Research Ethics Committee of Anhui University (Animal Ethics number: AHU3110) and Anhui Tianma National Nature Reserve, Anhui Province, China. Field experiments were approved by Anhui Tianma National Nature Reserve, Anhui Province, China.

Data collection

For the phylogenetic analyses, sequences of about half the species of Rhacophorus were used in combination with sequences of two outgroup species, Polypedates megacephalus (Rhacophoridae, Polypedates) and Spinomantis peraccae (Mantellidae, Spinomantis) (Li et al., 2013; Li et al., 2012a; Li et al., 2012b). Sequence data were obtained from GenBank (the GenBank Accession numbers are given in Table S1). In total, there were 149 individuals of 57 species of Rhacophorus involved (Fig. 1 and Table S1). All the taxonomic revisions within Rhacophorus were follow previous studies (Biju et al., 2013; Li et al., 2013; Orlov et al., 2012).

Figure 1 Sample sites of Rhacophorus species used in this study.

Sampling, DNA extraction, PCR amplification, and sequencing

Between 2012 to 2015, nine specimens of R. zhoukaiyae were collected from the Dabie Mountains, China (Pan et al., 2017). Muscle tissue from each individual was sampled and preserved in 100% ethanol for DNA extraction. Total DNA was extracted from the samples using a standard proteinase K/phenol-chloroform protocol (Sambrook, Fritsch & Maniatis, 1989). An EasyPure PCR Purification Kit (TransGene, Strasbourg, FR) was used to purify the DNA extractions. The sequences of 12S and 16S ribosome RNA (rRNA) of R. zhoukaiyae were collected from Pan et al. (2017). In addition, we also amplified and sequenced five nuclear gene fragments with the indicated primer pairs (Table S2), including brain-derived neurotrophic factor (BDNF), proopiomelanocortin (POMC), recombination activating gene 1 (RAG-1), rhodopsin exon 1 (RHOD) and tyrosinase exon 1 (TYR) (Bossuyt & Milinkovitch, 2000; Li et al., 2009; Van Der Meijden et al., 2007; Vieites, Min & Wake, 2007; Wiens et al., 2005). Polymerase chain reactions (PCR) were performed using a reaction mixture (25 µL) containing 1 µL genomic DNA (concentration 10–50 ng/µL), 2.5 µL 10× buffer, 1 µL of 2.5 mM MgSO4, 2 µL of 2 mM dNTPs, 1 U Taq polymerase (Meridian Bioscience, Singapore) and 0.3 mM of each of the primers. Pure molecular biology grade water was added to reach the appropriate volume. The amplification protocol included an initial denaturation step of 95 °C for 5 min; this was followed by 32 cycles of denaturation at 95 °C for 30 s, primer annealing at 51 °C –57 °C for 30 s, and an extension at 72 °C for 40 s–80 s, with a final extension at 72 °C for 10 min. PCR products were purified using an EasyPure PCR Purification Kit (TransGene) and sequenced using previous primers and the BigDye Terminator v3.0 Ready Reaction Cycle Sequencing Kit (Applied Biosystems, Foster City, CA, USA) following the manufacturer’s instructions on an ABI Prism 3730 automated sequencer. All the sequences obtained in this study were deposited into GenBank (Table S1). For the analyses, the sequences were trimmed to match data downloaded from GenBank, then all the sequences were aligned automatically using Clustal X version 1.83 (Thompson et al., 1997), followed by visual confirmation and manual adjustments. Nucleotide sites with ambiguous alignments were removed from the analyses, and gaps were analyzed as missing data.

Phylogenetic analyses

Two different datasets were generated for the different analyses. Dataset 1 was used for a phylogenetic analysis of Rhacophorus by Maximum Likelihood (ML) and Bayesian methods, and was comprised of the 12S and 16S rRNA gene together with the complete t-RNA for the valine sequence of the Rhacophorus species and the outgroups (Table S1). Dataset S2 contained more genes (12S, 16S, Val, BDNF, POMC, RAG-1, RHOD, TYR) of more individual and species than Dataset S1 (Table S1). However, it was only used to calculate a Bayesian consensus tree. The best-fit model of evolution was calculated by MrModeltest 1.0 b under the AIC criterion (Nylander, 2003). ML analyses were performed in RAxML version 8 (Stamatakis, 2014) and a general time reversible model of nucleotide substitution under the Gamma model of rate heterogeneity (i.e., GTRCAT). Support for the internal branches for the best-scoring tree was evaluated via the bootstrap test with 1,000 iterations. A Bayesian inference of phylogeny was performed using the MrBayes 3.1.2 software program (Huelsenbeck & Ronquist, 2005), using the best-fit substitution model. Two Markov Chain Monte Carlo (MCMC) models were run to provide additional confirmation of the convergence of posterior probability distributions. Analyses were run for 3,000,000 generations. Chains were sampled every 1,000 generations. The first 25% of the total trees were discarded as “burn-in” and the remaining trees were used to generate a majority-rule consensus tree and to calculate Bayesian posterior probabilities.

Divergence time analyses

To estimate divergence times of Rhacophorus, we applied a Bayesian MCMC method with mitochondrial genes (Dataset S1), which employs a relaxed molecular clock approach, as implemented in BEAST 1.7.4 (Drummond et al., 2012). We assumed a relaxed uncorrelated log normal model of lineage variation and a Yule Process prior to the branching rates based on the GTR + I + G model as recommended by MrModeltest 1.0 b (Nylander, 2003). Four replicates were run for 10,000,000 generations with tree and parameter sampling every 1,000 generations. The first 25% of samples were discarded as burn-in. All parameters were assessed by visual inspection using Tracer v. 1.5 (Rambaut & Drummond, 2007). The tree was generated and visualized with TreeAnnotator v. 1.6.1 (Rambaut & Drummond, 2010) and FigTree v. 1.3.1 (Rambaut, 2009), respectively. Calibration points were taken from Li et al. (2013) (Table 1). In addition, to visualizing the temporal accumulation of species, a log-transformed lineage-through-time (LTT) (Nee, May & Harvey, 1994) plot was constructed and compared with the null distribution for the LTT line simulated under the empirical pure-birth model. For visualizing diversification rate changes, we plotted the number of newly appearing species against the fixed time intervals of 2 million years (Ma) (Venditti, Meade & Pagel, 2010).

Table 1 Detailed results of molecular dating using BEAST 1.7.4, and the calibration points.

Labels for nodes correspond to Fig. 3. Unit: one million years. The abbreviation of time to most recent common ancestor is TMRC.

Node	TMRC	Mean (95%)	Mean(95%) (Li et al., 2013)	
Root	–	33.27 (25.11–40.20)	36.5 (31.2–40.9)	
a	Clade A, B, C	29.51 (25.34–34.07)	30.6 (25.2–34.7)	
b	Clade A, B	27.38 (22.44–32.17)	–	
c	Clade A	21.56 (17.92–25.22)	21.6 (17.5–25.1)	
d	Groups A2–A6	14.09 (10.96–17.41)	–	
e	Groups A3–A6	11.39 (8.89–14.16)	–	
f	Groups A4–A6	8.56 (6.43–10.88)	–	
g	Groups A5, A6	5.33 (3.92–6.99)	–	
h	Ggroup A6	2.9 (1.78–4.29)	–	
i	–	8.4 (6.43–10.29)	8.6 (5.5–9.8)	

Ancestral area reconstructions

Ancestral area reconstructions were inferred by the program RASP 3.2 (Yu et al., 2015) for speciational evolution in phylogenetic trees, using the Bayesian Binary MCMC (BBM) method (Ronquist & Huelsenbeck, 2003) and the statistical dispersal-vicariance method (S-DIVA) (Yu, Harris & He, 2010). To reconstruct ancestral areas on the basis of the topography, the distributional range of Asian Rhacophorus was divided into four regions, W, X, Y and Z (Fig. 2). W represents Southeast Asia, including the Indochinese Peninsula, Sundaland and the south margin of the Tibetan Plateau, X contains the Hengduan mountains and the mountains around the Sichuan Basin, Y refers to South China and Japan and Z represents India (Fig. 2). The tree data sets and the condensed tree were generated by BEAST 1.7.4 (Drummond et al., 2012). The distribution of each species was collected from http://maps.iucnredlist.org. For all analyses, the maximum number of ancestral areas at each node was constrained to three. The frequencies of an ancestral range at a node were averaged over all trees and each alternative ancestral range at a node was weighted by the frequency of occurrence for the node.

Figure 2 Chronogram and ancestral area reconstructions of Rhacophorus with outgroup species, Polypedates megacephalus and Spinomantis peraccae based on Dataset S1.

Branches in the tree are proportional to absolute ages (Ma). Node charts showed the relative probabilities of alternative ancestral distributions obtained by integrating the statistical dispersal-vicariance analysis (S-DIVA; above branches) and a Bayesian Binary MCMC method (BBM; below branches), and the first two areas with highest probability are shown corresponding to their relative probability on the area of one circle. Areas are divided for reconstructing ancestral areas. (W) Southeast Asia, including the Indochinese Peninsula, Sundaland, and the south margin of the Tibetan Plateau; (X) Hengduan mountains and the mountains around the Sichuan Basin; (Y) South China and Japan; (Z) India. (A) species mostly distributed in Southeast Asia and East Asia; (B) species distributed in Southeast Asia and India; (C) species distributed in Southeast Asia.

Results

Molecular phylogenetic analyses

The aligned mtDNA gene fragments from Rhacophorus consisted of 1,935 bp nucleotide positions before trimming (Dataset S1). After trimming, 1,851 nucleotide positions were retained for genealogical reconstructions. The fragments contained 934 constant and 917 potentially phylogenetically informative characters. The ML or BI phylogenetic approaches based on Dataset S1 resulted in virtually identical topology, and all terminal clades obtained relatively high-supporting values (Fig. S1). The genus Rhacophorus was supported as monophyletic containing four major clades (Fig. S1). For further probing of the dispersal process and diversification of the Asian tree frog, the molecular dating and ancestral area reconstructions were carried out. The phylogenetic tree, collected from the molecular dating, showed three distinct clades (A, B and C) in the genus of Rhacophorus (Fig. 2). There were some difference in the species distribution areas among the three clades. Species in clade A were mostly distributed in Southeast Asia and East Asia, species in Clade B were distributed in Southeast Asia and India, and species in lineage C only found in Southeast Asia. Clade A contained six groups, A1 to A6 (Fig. 2). The phylogenetic tree, based on Dataset S2, was largely consistent with the results from Dataset S1 (Fig. S2). However, there were some minor differences between them, such as the polyphyletic of clade B and C in Fig. S2 . But, generally, it did not affect the results of ancestral area reconstructions of Rhacophorus.

Molecular dating, ancestral area reconstructions and lineage through time

Dating analyses based on Dataset S1 suggested that the most recent common ancestor (MRCA) of Rhacophorus dates back to 29.51 Ma (median value; 95% of the highest posterior density [HPD] = 25.00–34.07 Ma) (Table 1 and Fig. 3).The MRCA of Clade A and Clade B was estimated at 27.38Ma (95% HPD = 22.44–32.17 Ma). The MRCA of Clade A was 21.56 Ma (95% HPD = 17.92–25.22 Ma) and the MRCA of Clade B was 26.73 Ma (95% HPD = 21.56–31.83 Ma).

Figure 3 Biogeographical history of Rhacophorus.

(A) Time-calibrated phylogeny of the genus Rhacophorus inferred from the mitochondrial dataset with an outgroup species, Polypedates megacephalus and Spinomantis peraccae. The light-blue bars through the nodes indicate 95% credibility intervals. Detailed time estimates for nodes with letter labels are given in Table 1; i corresponds to the clade A in Fig. 2; ii corresponds to clade B; iii corresponds to clade C; (B) climatic sequence of events including a global average delt 18O curve (right-hand axis) derived from benthic foraminifera which mirrors the major global temperature trends from the Paleocene to the Pleistocene (red line) (modified from Zachos et al. (2001), Zachos, Dickens &Zeebe (2008) and Favre et al., 2015). The establishment of ice sheets in the Northern Hemisphere is indicated by grey to black bars on top. The onset and development of the monsoon is symbolised by a blue polygon and its intensification by grey bars (I, II and III) (Wan et al., 2007; Jacques et al., 2011). The climate oscillations during the Quaternary are represented by a grey bar (IV) (Deng et al., 2011); (C) geological sequences of events are related to the diversification of Rhacophorus including the reconstructions historical land and sea in Southeast Asia and a graphical representation of the extent of the uplift of the TP through time. ① and ② show two Cenozoic reconstructions of land and sea in the Indo-Australian Archipelago (modified from Lohman et al., 2011). Red shading in ③ and ④ indicates the portion of the TP that had achieved altitudes comparable to the present day for each given time (modified from Mulch & Chamberlain (2006) and Favre et al., 2015).

Ancestral area reconstructions from S-DIVA and BBM analyses were largely similar with some minor differences (Fig. 2). All analyses supported Southeast Asia (Area W, Fig. 2) as the ancestral area of Rhacophorus and most speciation events were attributed to dispersal. The empirical LTT plot of Rhacophorus showed that, after a lengthy period of constant diversification, the diversification rate of the genus had increased during the middle Pliocene. The cumulative curve of species-birth per time interval showed that the diversification of Rhacophorus fluctuated through time, especially during 14–12 Ma and 10–4 Ma (Fig. 4).

Figure 4 Visualization of diversification rate shifts of Rhacophorus.

(A) Lineage-through-time plot (logarithmic scale) and 95% confidence intervals of lineage diversification; (B) cumulative curve of diversification rate per million years. The dashed line represents the period of rapid diversification in Rhacophorus.

Discussion

The dispersal process of Rhacophorus and its spread toward East Asia and India

Previous studies have indicated that the diversification of Rhacophoridae was closely linked to the India-Asia collision (57 Ma–35 Ma) (Li et al., 2013). Southeastern Asia houses three globally significant hot spots divided by sharp, yet porous biogeographic boundaries (Evans et al., 2003; Favre et al., 2015; Schmitt, Kitchener & How, 1995; Wallace, 1860). Studies have shown that the dynamics of the formation of biodiversity in Southeastern Asia is assumed to be interrelated with many geological events and a unique climatic history. Events such as the continuing processes of volcanic uplift and the emergence of many new islands in Indo-Australian Archipelago during the Miocene-Pliocene (Fig. 3C) (Esselstyn, Timm & Brown, 2009; Hall, 1996; Hall, 1998; Hall, 2002; Lohman et al., 2011), the rapid uplifts of the Tibetan Plateau (Shi et al., 1999), repeated sea level fluctuations during the Pleistocene (Bird, Taylor & Hunt, 2005; Esselstyn, Timm & Brown, 2009; Hall, 1998; Heaney, 1985; Heaney, 1986; Jansa, Barker & Heaney, 2006; Voris, 2000) and the onset of the Asian monsoon system (An et al., 2001; Qiang et al., 2001; Sun & Wang, 2005; Zhisheng et al., 2001). Many phylogeographical studies of plants and animals support this assumption (Deng et al., 2011; Klaus et al., 2016; Shi et al., 1999), such as those on Lilium (Gao et al., 2013), Delphinieae (Jabbour & Renner, 2012), Hyoscyameae (Tu et al., 2010), Mandragoreae (Tu et al., 2010), Saussurea (Wang et al., 2009), birds (Lei, Qu & Song, 2014; Tietze & Borthakur, 2012; Tietze et al., 2013; Yang, Dong & Lei, 2009), Hynobiidae (Zhang et al., 2006), lizards (Guo et al., 2011) and Spiny Frogs (Che et al., 2010), so the diversification and speciation in Rhacophorus may also be related to the special geological formations and the climatic history.

The phylogenetic analysis shows that Rhacophorus is composed of multiple lineages. In the phylogenetic tree with timescale, calculated by BEAST, Rhacophorus is composed of three major clades, A, B and C (Fig. 2). Among these clades, Clade C was the basal branch of Rhacophorus, which contained ten species from Southeast Asia, and the age of the MRCA of Rhacophorus was estimated at 29.51 Ma (i.e., 95% CI [25–34.07 Ma], Fig. 2 and Table 1). The MRCA of Clades B and Clade A was 27.38 Ma (95% CI [22.44 Ma–32.17 Ma]) during the Oligocene (Fig. 2, Table 1). The members of Clades A and B are mainly distributed in the south of the Tibetan Plateau margin, India and Eastern Asia (Fig. 1). Clade A contained six groups which were distributed in three areas: Southeast Asia (group A1), the south of the Tibetan Plateau margin (group A2) and an Eastern Asia (group A3 to A6) (Figs. 1 and 2). The MRCA of Clade A occurred 21.56 Ma ago (95% CI [17.92–25.22 Ma]; Fig. 2) and the time of the split of different groups was estimated at 14.09 Ma (A2 vs A3∼A6, 95% CI [10.96–17.41 Ma]), 11.39 Ma (A3 vs A4∼A6, 95% CI [8.89–14.16 Ma]), 8.56 Ma (A4 vs A5∼A6, 95% CI [6.43–10.88 Ma]) and 5.33 Ma (A5 vs A6, 95% CI [3.92–6.99 Ma]) respectively (Table 1). In addition, the LTT plot analysis indicated an increased diversification rate during two periods (14–12 Ma and 10–4 Ma) (Fig. 4). Basically, the above mentioned phylogeographical information reflected the trend of diversification and the speciation process. Obviously, the distribution of these species expanded continuously from Southern Asia to India and Eastern Asia, reaching as far as Japan (Fig. 2).

During the Oligocene and Miocene the uplift progressed, causing the extension of the Tibetan Plateau (Harrison et al., 1992; Mulch & Chamberlain, 2006). The start of the uplift of the northern Tibetan Plateau occurred at about 30 Ma BP (Sun & Wang, 2005) or slightly earlier (Wang et al., 2012b). Then, the eastern parts of the Tibetan Plateau likely reached an elevation comparable to the present-day elevation in the Mid to Late Miocene (from 15 to 5 Ma) (Axelrod, 1997; Currie, Rowley & Tabor, 2005; Jacques et al., 2011; Spicer et al., 2003; Tapponnier et al., 2001; Valdiya, 1999; Zhang et al., 2013). The southeastern edge of the Tibetan Plateau, the Hengduan mountain range, experienced rapid uplift only after the Miocene (5.33 Ma), reaching a peak elevation shortly before the Late Pliocene (5.33–2.66 Ma) (Li & Fang, 1999; Mulch & Chamberlain, 2006; Sun et al., 2011; Zheng et al., 2000), which separated several major rivers that ran in parallel (the Yangtze, Mekong, and Salween valleys) (Clark et al., 2004). This series of rapid Tibetan Plateau uplifts dramatically changed the terrain and landform in this area, which resulted in speciation, especially in animal groups (Che et al., 2010; Deng et al., 2011; Gao et al., 2013; Jabbour & Renner, 2012; Lei, Qu & Song, 2014; Li et al., 2013; Shi et al., 1999; Tietze & Borthakur, 2012; Tietze et al., 2013; Tu et al., 2010; Wang et al., 2009; Zhang et al., 2006). Zhang et al. (2006) found that the origin and phylogenetic divergence of the Hynobiidae had a correlation to the uplift of the Tibetan Plateau (Zhang et al., 2006). The phylogenetic history of Paini (Anura: Dicroglossidae) illuminates a critical aspect of the timing of geological events, especially for the uplift of the Tibetan Plateau (Che et al., 2010). On the other hand, the Tibetan Plateau and its adjacent mountain ranges acted as an orographic barrier to atmospheric circulation in Asia and consequently contributed to the formation of the Asian monsoon system, which was one of the major climatic changes in this region (Early Miocene, 24 Ma) due to the Tibetan Plateau’s considerable size and altitude (Guo et al., 2008; Kutzbach, Prell & Ruddiman, 1993; Liu & Yin, 2002; Ruddiman & Kutzbach, 1991; Song et al., 2010; Sun & Wang, 2005; Tang et al., 2013; Zhisheng et al., 2001). In the following millions of years, the East Asian monsoon intensified three times (∼15 Ma, ∼8 Ma and 4–3 Ma) (An et al., 2001; Jacques et al., 2011; Molnar, Boos & Battisti, 2010; Song et al., 2010; Sun & Wang, 2005; Valdiya, 1999; Wan et al., 2007; Zhisheng et al., 2001). The development of the Asian monsoon system directly gave birth to the warm and humid climate in the south of China (Sun & Wang, 2005), which was maybe favorable for the geographical spread and speciation of amphibians (Che et al., 2010; Thorn & Raffaelli, 2001; Wu et al., 2013; Zhang et al., 2006). In addition, the climate oscillations that began about 2.8 million years ago, in the Late Pliocene (Deng et al., 2011), also provided the chance for diversification and speciation of many species (Zhang, Fengquan & Jianmin, 2000), such as birds (Lei, Qu & Song, 2014), the Tibetan woolly rhino (Coelodonta thibetana) (Deng et al., 2011) and stream-dwelling frog (Feirana quadranus) (Wang et al., 2012a). Molecular dating suggested that the TMRC of Clade A and Clade B was during the Oligocene (22.44 Ma–32.17 Ma) (Fig. 3, Table 1). At same time, ancestral area reconstructions supported Southeast Asia (W) as the ancestral area of Rhacophorus and the dispersal events happened from ancestral area of Clade A and Clade B (Fig. 2, node b). In addition, the land and sea in the Indo-Australian Archipelago changed greatly during this period (Lohman et al., 2011), which may promote the dispersal events from Southeast Asia. In Clade A, the time of the split of subgroups was estimated from 14.09 to 5.33 Ma (Table 1). In addition, the time of most nodes in Clade B also occurred during this period (Fig. 3). Based on the LTT plot analysis, there were two increased diversification rate periods (14–12 Ma, 10–4 Ma) in Rhacophorus (Fig. 4). The series of Tibetan Plateau rapid uplifts (from 15 to 2.66 Ma) dramatically changed the landscape, which resulted in the diversification of species or speciation in this area (Che et al., 2010; Deng et al., 2011; Gao et al., 2013; Jabbour & Renner, 2012; Lei, Qu & Song, 2014; Li et al., 2013; Shi et al., 1999; Tietze & Borthakur, 2012; Tietze et al., 2013; Tu et al., 2010; Wang et al., 2009; Zhang et al., 2006) and the biotic interchange between the Indian subcontinent and mainland Asia (Klaus et al., 2016). In addition, the intensified East Asian monsoon (∼15 Ma, ∼8 Ma and 4–3 Ma) directly gave birth to the warm and humid climate in the south of China, which was favorable for the geographical spread and speciation of amphibians (Che et al., 2010; Thorn & Raffaelli, 2001; Wu et al., 2013; Zhang et al., 2006). Obviously, the diversification events in Rhacophorus were in line with the time frame of the orogenic movement and climatic histories, especially the staged rapid uplift of the Tibetan Plateau and the enhanced Asian monsoon system (Figs. 2 and 3). Therefore, we think that the diversification and speciation events in Clade A and Clade B, are related to the staged uplift of the Tibetan Plateau and the subsequent chain-reaction events, such as the establishment of the Asian monsoon system, which facilitated the radiations and speciation of amphibians (Che et al., 2010; Thorn & Raffaelli, 2001; Wu et al., 2013; Zhang et al., 2006).

Overall, the evolutionary history of Rhacophorus originated approx 30 Ma Bp (Oligocene). Basically, it is the dispersal process from its ancestral area, Southeast Asia, toward India and East Asia. During the process, Rhacophorus diversified by multiple factors, such as geographic shifts, the staged rapid rises of the Tibetan Plateau with parallel climatic changes, the reinforcement of the Asian monsoons (15 Ma, 8 Ma and 4–3 Ma) and alternating glacial-interglacial oscillations.

Supplemental Information

Figure S1 Bayesian inference tree of the genus Rhacophorus based on mitochondrial data (Dataset S1)

The nodal numbers are posterior probabilities ( >80% retained) and ML ( >50% retained).

Click here for additional data file.

Figure S2 Bayesian phylogenetic tree of the genus Rhacophorus based on the combined nuclear and mitochondrial dataset (Dataset S2) with posterior probabilities for branches ( >50% retained)

Group B and Group C were correspongding to the Clade B and Clade C in Fig. 2, respectively.

Click here for additional data file.

Table S1 Samples, with sampling site, museum voucher nos., and GenBank accession nos. of corresponding sequences

“—” represents no molecular data.

Click here for additional data file.

Table S2 Primers used in PCR and sequencing

Click here for additional data file.

Data S1 DNA Sequencing

The raw DNA sequence data of R. zhoukaiyae.

Click here for additional data file.

We thank Wenliang Zhou, Zhonglou Sun, Zhaojie Peng and Xiaonan Sun for their help in sample collecting, John Bailey (Department of Genetics, University of Leicester) and Martin Burrows for input concerning the quality of the writing as regards the English language, Jiatang Li (Chengdu Institute of Biology, Chinese Academy of Sciences) for the study design, and thank the reviewers (Nguyen Tao, Gururaja Kotambylu Vasudeva and an anonymous reviewer) for their suggestions. We thank Tianma National Nature Reserve in Anhui Province for the investigation project and the sample collection.

Additional Information and Declarations

Competing Interests

Author Contributions

Animal Ethics

Field Study Permissions

Data Availability

The authors declare there are no competing interests.

Tao Pan conceived and designed the experiments, performed the experiments, analyzed the data, wrote the paper, prepared figures and/or tables, reviewed drafts of the paper.

Yanan Zhang conceived and designed the experiments, performed the experiments, analyzed the data, wrote the paper, reviewed drafts of the paper.

Hui Wang performed the experiments, analyzed the data.

Jun Wu prepared figures and/or tables.

Xing Kang performed the experiments, contributed reagents/materials/analysis tools.

Lifu Qian conceived and designed the experiments, performed the experiments, contributed reagents/materials/analysis tools, prepared figures and/or tables.

Jinyun Chen performed the experiments.

Dingqi Rao and Jianping Jiang analyzed the data, reviewed drafts of the paper.

Baowei Zhang conceived and designed the experiments, analyzed the data, wrote the paper, reviewed drafts of the paper.

The following information was supplied relating to ethical approvals (i.e., approving body and any reference numbers):

This investigation project and the sample collection were approved by Anhui Tianma National Nature Reserve, Anhui Province, China. Animal Ethics number: AHU3110.

The following information was supplied relating to field study approvals (i.e., approving body and any reference numbers):

Field experiments were approved by Anhui Tianma National Nature Reserve, Anhui Province, China.

The following information was supplied regarding data availability:

NCBI:

KU601449 –KU601493.

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
