# Peer review of "The reanalysis of biogeography of the Asian tree frog, Rhacophorus (Anura: Rhacophoridae): geographic shifts and climatic change influenced the dispersal process and diversification"

_PeerJ, doi:10.7717/peerj.3995_

## Round 0.1 · original submission · Major Revisions

Although all three reviews were brief, the reviewers found several strong points in your manuscript but all 3 identified aspects that need to be addressed before it would be acceptable for publication.

·

Basic reporting

No comment

Experimental design

No comment

Validity of the findings

Orlov, Poyarkov, Vassilieva, Ananjeva, Nguyen, Sang, and Geissler, 2012, Russ. J. Herpetol., 19: 23-64 noted extensive confusion with Rhacophorus robertingeri in the literature and mentioned R. chuyangsinesis considered as a junior synonym of R. calcaneus. Thus, the authors need to correct data.

Additional comments

Because computer programs change over time and the default parameters are not necessarily rigid, it would help to specify what the default parameters are. This also helps the reader to know that you judged particular parameters and associated assumptions to be the best ones for your data.

·

Basic reporting

No comment

Experimental design

No comment

Validity of the findings

No comment

Additional comments

Excellent manuscript by the authors. Since the manuscript talks about geological phenomenon, I felt Two important papers from Indian sub-continent were mission from the manuscript. I would be good if the following papers are cited in the revised manuscript.
1. Taxonomic revisions of Rhacophorus genus in Indian sub-continent was given in Biju et al 2013. This paper did not find a place in the manuscript.
2. Similarly, Vijayakumar et al 2016 provided insights on glaciation, gradient and geography of Western Ghats, that perhaps shaped diversification in Raorchestes genus. This can be cited in the manuscript.

Reviewer 3 ·

Basic reporting

Format of text is not consistent throughout the manuscript. For example, "et al." are italicized in some case, but not, in normal, in other case. Correct such basic formatting throughout the manuscript before submission.

Experimental design

The results of the present study and the former one (Li et al., 2013) were very different. Especially, the estimated ancestral area is quite different. The present authors must clarify any difference in samples, analysis, and interpretation on the results between them.

Validity of the findings

As shown on Fig.S1, 1) a lot of nodes lacked a significant supports. Further, 2) the topology is not consistent between Fig. S1 and Fig. 2. Need to improve or explain on these two major concerns.

---

## Round 0.2 · accepted · Accept

All three reviewers were unanimous in recognising that you did an excellent job addressing their concerns.

·

Basic reporting

No comment

Experimental design

No comment

Validity of the findings

No comment

Additional comments

Excellent work by the authors. Congrats!

·

Basic reporting

no comment

Experimental design

no comment

Validity of the findings

no comment

Additional comments

Thanks for addressing the comments.

Reviewer 3 ·

Basic reporting

I confirmed the improved MS.

Experimental design

No problem.

Validity of the findings

OK.

Additional comments

I am looking forward for seeing this publication.